

# A ground motion prediction model for the Italian region based on a mixture of experts framework

Jinfeng Dai[1,2], Zifa Wang[1,2], Dengke Zhao[1,2], Xiangying Wang[3], Jianming Wang[1,2], Zhaoyan Li[1,2], Zhaodong Wang[1,2], Jintao Xiao[1,2]

[1]Key Laboratory of Earthquake Engineering and Engineering Vibration, Institute of Engineering Mechanics, China Earthquake Administration, Harbin, 150080, China
[2]Key Laboratory of Earthquake Disaster Mitigation, Ministry of Emergency Management, Harbin, 150080, China
[3]School of Architecture and Civil Engineering, Huangshan University, Huangshan, 245041, China

*Correspondence to*: Zifa Wang (zifa@iem.ac.cn)

**Abstract.** Earthquake ground-motion prediction is crucial for seismic design, seismic hazard assessment, and the resilience of urban infrastructure. Although extensive research has been conducted for Italy, existing models cover only a limited range of earthquake types, exhibit insufficient accuracy and uncertainty control under complex scenarios – thus lowering reliability – and provide a restricted set of ground-motion intensity measures (IMs) that cannot meet the multi-indicator needs of engineering practice and risk assessment. To address these issues, this study proposes a ground-motion prediction model for

Italy based on a Mixture of experts (MOE) framework, in which XGBoost is employed as the expert submodels to enhance predictive accuracy and stability across diverse scenarios. We conduct a systematic comparison between the proposed MOE-XGB and a baseline Gaussian process regression model with an exponential kernel (GPR, exponential). The results show stable and balanced improvements across multiple IMs – such as peak ground acceleration (PGA), peak ground velocity (PGV), and spectral acceleration (SA) at different periods – demonstrating advantages in both accuracy and robustness.

Furthermore, using the larger and more diverse ITACA (Italian Accelerometric Archive) dataset, we retrain and evaluate MOE-XGB. The model achieves higher accuracy on all considered metrics and maintains stable performance in generalization tests based on independent earthquake events, highlighting strong generalization capability and robustness. In summary, the proposed MOE-XGB provides a high-accuracy and broadly applicable solution for ground-motion prediction in Italy; meanwhile, the framework exhibits good transferability and scalability, offering a useful reference for fusion-model-

driven ground-motion prediction in Europe and other regions.

## 1 Introduction

Italy experiences approximately 40 000 earthquakes annually, with seismic activity affecting over 21 million residents in high-risk areas across the peninsula. The 2016 central Italy earthquake sequence caused nearly 300 deaths and billions of dollars in losses, highlighting the critical role of precise ground-motion prediction in hazard assessment and risk mitigation.

With the advancement of urbanization in seismically active zones and the aging of infrastructure, the limitations of existing



ground motion prediction equations (GMPEs) and ground motion prediction models (GMPMs) have become increasingly apparent.

GMPEs and GMPMs constitute core tools for modern seismic hazard assessment, providing support for earthquake engineering, emergency preparedness, and urban planning. Historically, empirical models have established fundamental
relationships between source parameters, propagation paths, and site conditions. The NGA-West project in the United States initially proposed five attenuation relations (Abrahamson and Silva, 2008; Boore and Atkinson, 2008; Campbell and Bozorgnia, 2008; Chiou and Youngs, 2008; Idriss, 2008), establishing a benchmark for subsequent research. Based on Italian strong-motion data, Scasserra et al. (2009) noted that while Italian data show consistency with NGA GMPEs regarding magnitude and site shear-wave velocity scaling, short-period ground motions attenuate faster; thus, adjustments to the
constant term, distance attenuation slope, and fictitious depth term were necessary to improve applicability. Building on this, Bindi et al. (2011) developed a set of GMPEs for the Italian strong-motion database, covering magnitudes 4.0 to 6.9 and distances up to 200 km for horizontal and vertical ground-motion parameters, with standard deviations between 0.34 and 0.38. However, this model exhibited large variability in predicting small-to-moderate magnitude events, indicating its limitations. To further improve the model, Lanzano et al. (2019) revised the ITA10 ground-motion prediction model,
extending the applicable magnitude range to above 6.9 and vibration periods to 10 s, using RotD50 as the ground-motion parameter. They introduced fault distance and heteroscedastic variability models, significantly improving the accuracy of near-field strong-motion predictions while reducing the standard deviation at intermediate-to-long periods by approximately 20 %. On this basis, Michelini et al. (2019) utilized the GMPE of Bindi et al. (2011), combined with data from the National Institute of Geophysics and Volcanology (INGV), to develop post-earthquake damage distribution maps, providing
important tools for post-disaster assessment. In recent years, the introduction of machine learning has opened new possibilities for ground-motion prediction, as it can capture nonlinear relationships and complex interactions that are difficult to model with traditional parametric methods. Mori et al. (2021) employed Gaussian process regression (GPR), combining approximately 16 000 accelerometer data points and 46 000 geological and geophysical data points, and considering stratigraphic and geomorphological conditions, to generate ground-motion prediction maps with a resolution of 50 m × 50 m.
This method significantly improved the precision and accuracy of near-real-time prediction; notably, the GPR model outperformed traditional GMPEs when accounting for the significant influence of topographic features and local site conditions on ground-motion amplification patterns.

Although ground-motion prediction models designed specifically for Italy have been widely applied (Bindi et al., 2006; Bindi et al., 2014), existing methods still possess significant limitations (Brando et al., 2020; Falcone et al., 2020a). First,
these models are typically developed for a single earthquake type (mainly shallow crustal events), making it difficult to fully adapt to diverse source types and complex geological conditions (Agrawal and McCloskey, 2024). This restricts the ability of models to characterize the diversity of Italian seismic activity and regions where multiple source types jointly constitute the overall seismic risk, thereby affecting the comprehensive assessment of seismic hazard. Second, prediction accuracy is insufficient; particularly in complex scenarios involving topographic amplification, site-specific effects, and different source





mechanisms, the models exhibit high variability, limiting their reliability. Furthermore, single-model approaches cannot fully leverage the complementary advantages of different modeling techniques, potentially missing opportunities to enhance prediction performance through ensemble methods or expert system architectures. Existing models are also limited in the scope of their output parameters. For example, the machine learning model of Mori et al. (2022) predicts only limited ground-motion parameters such as peak ground acceleration (PGA), peak ground velocity (PGV), spectral acceleration at 0.3

s (SA $T$=0.3), spectral acceleration at 1 s (SA $T$=1), and spectral acceleration at 3 s (SA $T$=3), failing to provide comprehensive ground-motion prediction values. This is particularly problematic for comprehensive hazard analyses, which require determining multiple ground-motion parameters for different structural types and performance objectives; a single output parameter cannot meet the diverse needs of engineering applications and risk assessment frameworks. Additionally, due to regional differences in seismic wave propagation, site effects, and source characteristics, models developed based on a

specific tectonic environment may fail in other regions, limiting the applicability and robustness of prediction results. This lack of validation across different tectonic environments and source characteristics further raises concerns regarding model transferability and reliability in international applications. In summary, future research needs to integrate multi-source data, expand the range of prediction parameters, and develop more universal and robust models to enhance the comprehensiveness and accuracy of ground-motion prediction.

This paper proposes a novel mixture of experts (MOE) framework, employing XGBoost as the expert model, aimed at addressing the fundamental limitations of existing ground-motion prediction models and providing comprehensive predictions for ground motion in Italy. We designed a unified modeling approach capable of adapting to multiple earthquake types and achieving prediction accuracy superior to traditional single models, thereby providing more reliable support for seismic hazard assessment. The mixture of experts model effectively captures the unique characteristics of earthquakes of

different magnitudes by training specialized expert models based on earthquake data subsets partitioned by magnitude. To determine the optimal model configuration, we conducted systematic testing on the number of expert models, and the results indicated that employing three expert models performed best across multiple evaluation metrics. Specifically, we divided the optimized dataset into three categories – small, medium, and large – based on magnitude, and performed Bayesian optimization for each category to construct expert models targeting different magnitude ranges. This hierarchical

optimization strategy enables the model to better adapt to magnitude-dependent ground-motion characteristics, thereby significantly improving prediction accuracy across the range of Italian seismic activity.

The main contributions of this study are as follows:

(1) Development and validation of the MOE-XGB framework: A ground-motion prediction model based on a mixture of experts (MOE) framework is proposed, incorporating XGBoost as the expert submodel. Through comparison with the

existing GPR model, the significant accuracy improvement of this model across multiple ground-motion intensity measures is validated.

(2) Application in multi-type earthquake scenarios: Based on the ITACA dataset, which has a larger sample size and broader coverage of earthquake types, the MOE-XGB model was retrained and evaluated. The results verify the high-precision



prediction capability of the model across diverse seismic scenarios. Comparisons with the existing GPR model and the
traditional Bindi 2017 model demonstrate more significant improvements in accuracy.

(3) Independent testing and generalization capability validation: Through testing on independent earthquake events, it is
further proven that the MOE-XGB model maintains stable prediction performance when facing unknown earthquake events,
reflecting its strong generalization capability and robustness.

The significance of the development of the MOE-XGB model is reflected not only in the improvement of prediction
accuracy but also in the innovation of seismic hazard modeling methods, the deep understanding of magnitude-dependent
prediction strategies, and the practical application in Italian seismic risk assessment. By overcoming the limitations of
existing methods, this study aims to lay the foundation for the next generation of ground-motion prediction models and to
develop a more universal prediction framework applicable to diverse seismic environments. In summary, the proposed
framework exhibits good transferability and scalability, offering a useful reference for fusion-model-driven ground-motion
prediction in Europe and other regions.

## 2 Data sources and overview

The ITACA dataset is a crucial resource for ground-motion prediction research in Italy, containing 52 706 records covering
2338 seismic events between 1972 and 2022, recorded by 2294 stations (Foti et al., 2011). This dataset integrates core data
from the ITACA database and supplements small-magnitude velocity records from under-sampled regions in Italy (e.g.
Liguria, Piedmont, and western Sicily), while also incorporating network records from neighboring countries (e.g. France,
Switzerland, Slovenia, Albania, and Montenegro) to improve the spatial and azimuthal coverage of events. This study
utilizes the SA flatfile, which contains the median spectral acceleration (RotD50) of the horizontal components (5 %
damping) and relevant metadata. RotD50 represents the median spectral acceleration after the rotation of horizontal
components; it is calculated by rotating the two horizontal components (north–south and east–west) of the seismic record,
computing the resultant spectral acceleration at all possible angles, and taking the 50th percentile. It characterizes direction-
independent ground-motion properties and is widely applied in earthquake engineering and ground-motion prediction
models.

The MORI dataset, compiled by Mori et al., includes seismological parameters, geophysical data, and morphological data,
which are used for the training of machine learning models and ground-motion prediction. The seismological parameters
consist of 15 779 data points, including PGA, PGV, and SA at 0.3, 1.0, and 3.0 s. These data are obtained from Italian (ESM,
ITACA) and European databases, covering parameters such as moment magnitude ($M$), epicentral distance ($R$), and
hypocentral depth ($H$), with a focus on shallow crustal events ($H < 35$ km). The geophysical data characterize site conditions
using $V_{S30}$ (the time-averaged shear-wave velocity in the top 30 m), combining field tests with the $V_{S30}$ map from Mori et al.
(2020a) as supplementary data Luzi et al. (2016, 2020). The morphological data include elevation ($h$) obtained from the



ALOS World 3D-30m digital elevation model (DEM) and terrain gradient parameters ($h_x$, $h_y$, $h_{xx}$, $h_{yy}$) generated via GIS ALOS (2021).

Figure 1 clearly shows that the ITACA dataset significantly outperforms the MORI dataset in terms of data density and comprehensiveness. The ITACA dataset covers a wider range of source depths and magnitudes, and the data volume far exceeds that of the MORI dataset. Therefore, we select the ITACA dataset as the primary data source for developing a

ground-motion prediction model applicable to the Italian region to ensure the robustness and applicability of the model. Meanwhile, due to its unique data distribution characteristics, the MORI dataset serves as an independent test dataset for verifying model performance.

Data processing: To reduce data skewness and enhance model stability, a base-10 logarithmic transformation is applied to the PGA and SA parameters. Additionally, ground-motion records containing missing values are removed to ensure the

completeness and reliability of the dataset. Dataset splitting: After screening, 45 080 valid records are retained. These are divided into a training set (31 554 records), a validation set (6763 records), and a test set (6763 records) according to a ratio of 70 %, 15 %, and 15 %, respectively, to support model training, optimization, and performance evaluation.





Figure 1. Comparison of data distribution for the ITACA (blue) and MORI (red) datasets.

## 3 Model

The mixture of experts (MOE) model is a machine learning technique that collaboratively processes complex tasks through multiple specialized "expert" models; it has performed exceptionally well in the fields of natural language processing and computer vision in recent years. Given the nonlinearity of ground-motion data and its spatial heterogeneity, this study introduces the MOE framework into ground-motion prediction, aiming to enhance prediction accuracy and stability.





### 3.1 Model overview

The MOE architecture was first proposed by Jacobs et al. (1991) and has recently been fully utilized in the open-source large language model DeepSeek. DeepSeek (DeepSeek-AI et al., 2024; Shao et al., 2024) utilizes MOE to achieve efficient training and inference, enhancing performance and scalability. Its core concept is "divide and conquer", where a complex task is decomposed into subtasks processed by specialized expert models, and the results are then integrated through a gating mechanism. In earthquake engineering, ground-motion data possess high nonlinearity and spatial heterogeneity, making it 155 difficult for traditional single models to comprehensively capture these characteristics. This study innovatively applies the MOE framework to ground-motion prediction, employing multiple expert models to capture multi-scale features and dynamically selecting and combining experts through a gating network. This design enables the model to adaptively handle seismic events with different tectonic environments, magnitudes, and propagation paths, providing a new paradigm for ground-motion prediction.

To achieve precise prediction of PGA and SA, this study proposes the MOE-XGB model, which combines the advantages of the XGBoost algorithm and neural networks, as shown in Figure 2. The MOE-XGB model enhances predictive performance through collaborative division of labor, resolving the difficulty single models face in comprehensively capturing all information. The model consists of two main parts: expert models and a gating network. Through ablation experiments comparing the performance of 1, 2, 3, and 4 expert models, it was found that the test results of 3 expert models performed 165 best across various metrics. Comprehensively considering prediction accuracy and computational efficiency, the number of expert models was finally determined to be 3. The expert models are based on the XGBoost algorithm, which excels at processing structured data such as earthquake magnitude, focal depth, and $V_{s30}$. This study assigns three XGBoost expert models to each SA period, employing three different parameter settings to capture diverse data features, ensuring adaptation to various seismic scenarios. The gating network is a compact neural network that assigns weights to each expert based on 170 input earthquake features. Its processing procedure involves extracting feature representations $g(x)$ through multi-layer linear transformations and non-linear activation functions (ReLU), and then calculating the weights $w(x)$ using the Softmax function. The final prediction is the weighted sum of the expert outputs, expressed as the base-10 logarithm of the predicted value $\log_{10} \hat{y}$. The specific implementation is shown in Eqs. (1), (2), and (3):

$$g(x) = Linear_3\left(ReLU\left(Linear_2\left(ReLU(Linear_1 X)\right)\right)\right) \tag{1}$$

$$w(x) = Soft\max\left(g(x)\right) \tag{2}$$

$$\log_{10} \hat{y} = \sum_{k=1}^{K}\left[w\left(g(x)\right)_k \cdot f_k(x)\right] \tag{3}$$

where: $Linear_1$ transforms the input dimension to 64 dimensions; $Linear_2$ transforms 64 dimensions to 32 dimensions; $Linear_3$ transforms 32 dimensions to the number of experts; $x$ represents the input feature vector; $g(x)$ represents the latent feature representation; $w_k(x)$ represents the weight assigned to the experts by the gating network; $f_k(x)$ represents the



prediction result of the $k$-th expert model; $K$ is the number of expert models; and $\hat{y}$ represents the predicted value of the ground-motion parameter.

This study selects 10 feature parameters, including moment magnitude, hypocentral depth, epicentral distance, $V_{s30}$, station elevation, and the latitude and longitude information of both the earthquake and the station (Chen et al., 2022). These features help the model capture the physical and spatial characteristics of earthquakes. The dataset is divided into a training

set (70 %), a validation set (15 %), and a test set (15 %). Model performance is evaluated using the root mean square error (RMSE), correlation coefficient (R), and standard deviation of prediction residuals (Std Dev), reflecting accuracy, correlation, and stability, respectively.

## 3.2 Bayesian optimization

To enhance the performance of the MOE-XGB model in ground-motion spectral acceleration prediction, this study adopts

Bayesian optimization to determine optimal hyperparameters for each expert model, with the objective of minimizing the mean squared error (MSE) of the validation set. The goal is to generate a unique set of optimal hyperparameters for the XGBoost expert models at each SA period, ensuring parameter diversity through different dataset partitions.

The optimization process begins by loading the ground-motion dataset, which is divided into a training set (70 %) and a validation set (30 %), containing 10 features (e.g. moment magnitude, hypocentral depth, and epicentral distance). To

generate hyperparameters targeting different seismic scenarios, the training and validation sets are classified by magnitude into three parts: small earthquakes ($M < 4$), moderate earthquakes ($4 \leq M < 6$), and large earthquakes ($M \geq 6$). These sub-datasets are combined to form three groups of optimization datasets, targeting small, moderate, and large earthquake scenarios, respectively, to generate corresponding optimal hyperparameters. Bayesian optimization is conducted independently for each SA period, using the validation set MSE as the objective function, and calculating hyperparameter

performance by training and evaluating the XGBoost model. The entire process uses only the training and validation sets, without the test set, thereby avoiding the risk of data leakage. Each optimization iteration explores the hyperparameter space (e.g. tree depth, learning rate, and sampling ratio) to ensure that the three expert models obtain unique hyperparameter configurations, enhancing the model's adaptability and prediction accuracy for different seismic scenarios.






Figure 2. Schematic diagram of the MOE-XGB model structure.

## 4 Results

### 4.1 Model evaluation framework

Based on the two earthquake ground-motion datasets, this study conducted two comprehensive tests on the model. The datasets were divided according to a training: validation: testing ratio of 70:15:15, ensuring the robustness of model development and the reliability of evaluation. Optimal hyperparameters were determined through Bayesian optimization, and





the MOE-XGB model was employed for training, validation, and testing. Performance analysis focuses on residual distribution; the residual is defined as the difference between the base-10 logarithms of the predicted value and the actual value, as shown in Eq. (4):

$$\delta = \log_{10}(\hat{y}) - \log_{10}(y) \tag{4}$$

where $\hat{y}$ represents the predicted value based on input features; $y$ represents the actual observed data value; $log_{10}(\cdot)$ denotes

the base-10 logarithm function; and $\delta$ represents the residual, indicating the deviation between the predicted and true values on a logarithmic scale.

To conduct an in-depth analysis, decomposing residuals into inter-event and intra-event components holds significant theoretical and practical value. This decomposition originates from the statistical method of mixed-effects models and allows for a more comprehensive evaluation of ground-motion prediction model performance. Therefore, adopting the method

proposed by Abrahamson and Youngs (1992), the relationship between the observed and predicted ground motions is modeled by decomposing the total residual into inter-event and intra-event components, as shown in Eq. (5):

$$\log_{10} y_{ij} = \log_{10} \hat{y}_{ij} + \eta_i + \varepsilon_{ij} \tag{5}$$

where $y_{ij}$ represents the observed value of the $j$-th record for the $i$-th event; $\hat{y}_{ij}$ represents the predicted value given by the model; $\eta_i$ denotes the inter-event residual of the $i$-th event; and $\varepsilon_{ij}$ denotes the intra-event residual.

To evaluate the prediction accuracy and robustness of the MOE-XGB model, this study analyzed the relationship between residuals and moment magnitude, epicentral distance, intra-event residuals, and inter-event residuals, as well as the residual probability density distribution. Evaluation metrics include the root mean square error (RMSE), correlation coefficient (R), and standard deviation ($\sigma$), which quantify prediction error, trend capturing capability, error dispersion, prediction accuracy, and model goodness-of-fit, respectively. The residual analysis results indicate that the MOE-XGB model performs

excellently in capturing the complex nonlinear relationships of earthquake ground-motion data, verifying its reliability in seismic hazard assessment and earthquake engineering applications.

## 4.2 Comparison of model performance

This subsection utilizes the MORI dataset compiled by Mori et al. (containing 15 779 seismological data points, covering seismological parameters such as moment magnitude (*M*), hypocentra depth (*H*), epicentral distance (*R*), peak ground

acceleration (PGA), peak ground velocity (PGV), and spectral acceleration (SA); geophysical data such as; and morphological data such as terrain gradient and station elevation; focusing on shallow active crustal regions with *H* < 35 km; $V_{s30}$ data are integrated from approximately 11 300 field tests and the $V_{s30}$ map of Mori et al. (2020a)) to validate the performance of the MOE-XGB model. To ensure a fair comparison, the GPR (exponential kernel) model adopted by Mori et al. is selected as the benchmark, as it was trained and predicted based on the same dataset. By comparing the performance of

the MOE-XGB and GPR models on the same dataset, we evaluate the superiority of MOE-XGB in capturing the complex nonlinear relationships of earthquake ground motion, highlighting its performance advantages.





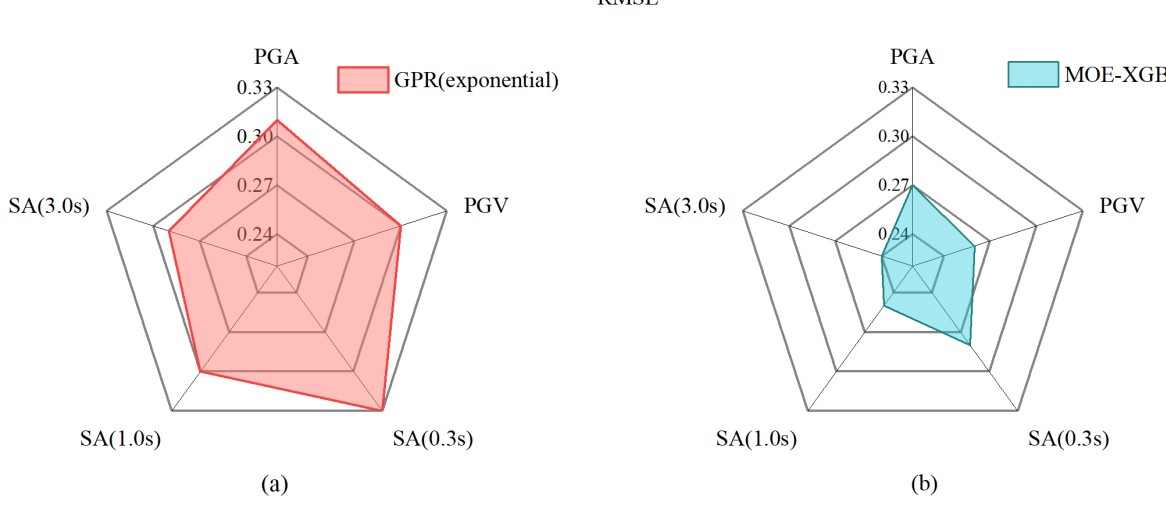

Figure 3. Comparison of RMSE between the GPR (exponential) model and the MOE-XGB model across different ground-motion parameters.

As shown in Figure 3 and Table 1, compared with the GPR model using an exponential kernel function, the proposed MOE-
XGB model exhibits lower RMSE across all ground-motion parameters. Specifically, MOE-XGB reduced the RMSE by 12.9 % for PGA, 13.3 % for PGV, 15.2 % for SA (0.3 s), 16.7 % for SA (1.0 s), and 17.2 % for SA (3.0 s).

Table 1. RMSE performance of different models.

| Model | PGA (Gal) | PGV (cm s$^{-1}$) | SA ($T$=0.3 s) (Gal) | SA ($T$=1.0 s) (Gal) | SA ($T$=3.0 s) (Gal) |
|---|---|---|---|---|---|
| GPR (exponential) | 0.312 | 0.303 | 0.331 | 0.305 | 0.293 |
| MOE-XGB | 0.273 | 0.264 | 0.286 | 0.257 | 0.249 |

To compare the residual performance of the MOE-XGB and GPR (exponential) models in magnitude–distance space under a unified standard, we plotted residual comparison figures. Figure 4 displays magnitude on the horizontal axis, and Figure 5
displays epicentral distance on the horizontal axis; each subplot corresponds to the residuals of different prediction metrics, including PGA, PGV, SA (0.3 s), SA (1.0 s), and SA (3.0 s). In the figures, green scatter points represent the prediction residuals of MOE-XGB, and red scatter points represent the prediction residuals of GPR. In this way, we demonstrate the spatial distribution characteristics of different models in ground-motion prediction from multiple dimensions, facilitating a more intuitive comparison of their predictive performance under different magnitude and source distance conditions.
In the overall statistical results of the five prediction metrics (PGA, PGV, SA (0.3 s), SA (1.0 s), and SA (3.0 s)), MOE-XGB and GPR (exponential) exhibit significant differences in residual distribution. For residuals with an absolute value greater than 1 ($|\delta| > 1$), the proportions corresponding to the five prediction metrics of MOE-XGB are 0.17 %, 0.13 %, 0.21 %, 0.08 %, and 0.08 %, respectively, with an overall average of only 0.13 %; whereas the corresponding proportions for GPR





are 1.27 %, 0.51 %, 0.97 %, 0.59 %, and 0.68 %, respectively, with an overall average of 0.80 %. Compared to GPR, MOE-
XGB reduces large residual points by approximately 83.8 %, demonstrating more robust predictive capability within the magnitude–distance space.

In the statistics for residuals with an absolute value greater than 0.5 ($|\delta| > 0.5$), the residual proportions for MOE-XGB are 2.95 %, 2.45 %, 3.80 %, 2.99 %, and 2.53 %, respectively, with an overall average of 2.94 %; whereas the corresponding proportions for GPR are 9.66 %, 7.79 %, 9.97 %, 7.46 %, and 7.09 %, respectively, with an overall average of 8.39 %.
Evidently, MOE-XGB also possesses significant advantages in controlling moderate-amplitude residuals, reducing residual points by approximately 65.0 % compared to GPR.

In summary, MOE-XGB exhibits a more concentrated residual distribution, with the majority of residuals concentrated in the $|\delta| < 0.5$ interval, significantly outperforming GPR. Combined with the graphical results, it can be further observed that the residuals of MOE-XGB are distributed more tightly around the $\delta = 0$ axis overall, while the residuals of GPR present a wider
distribution range and a higher proportion of large residual points. This fully demonstrates that MOE-XGB possesses more prominent advantages in both the accuracy and stability of ground-motion prediction.

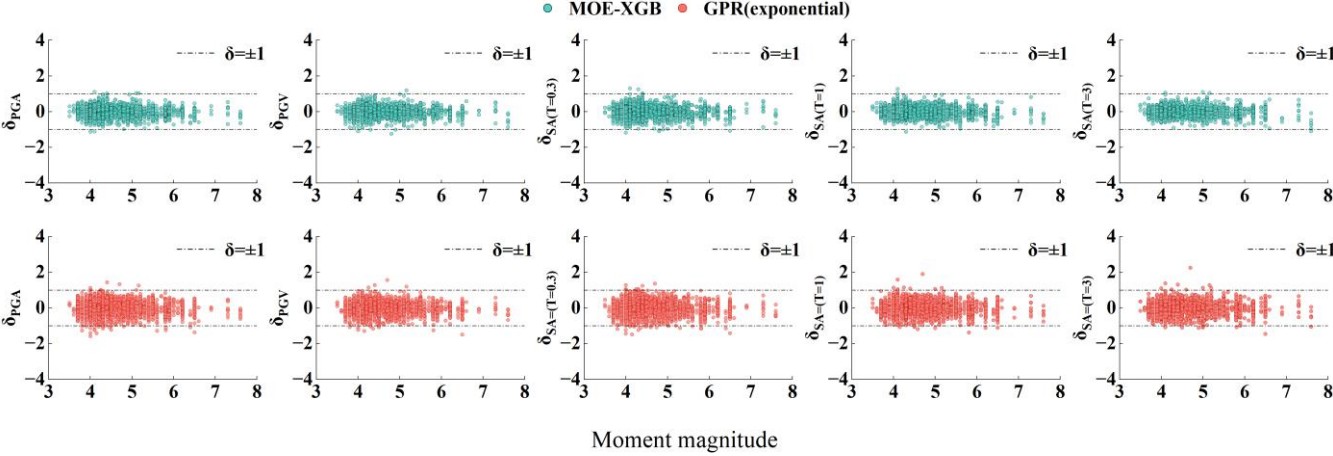

Figure 4. Variation of residuals with magnitude for the MOE-XGB and GPR (exponential) models based on the MORI dataset for PGA and different SA periods.




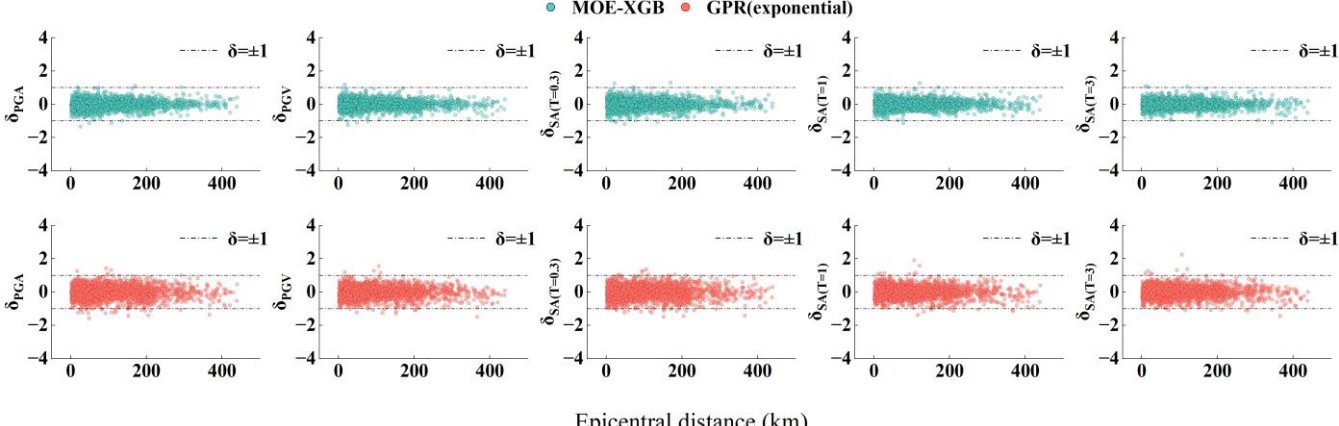

Figure 5. Variation of residuals with epicentral distance for the MOE-XGB and GPR (exponential) models based on the MORI dataset for PGA and different SA periods.

To comprehensively evaluate the prediction residual characteristics of the MOE-XGB and GPR (exponential) models under five ground-motion intensity measures (PGA, PGV, and spectral acceleration at $T = 0.3$ s, 1.0 s, and 3.0 s), this paper selected two statistical metrics for comparative analysis: peak density and peak position. Among them, peak density reflects the degree of concentration of residuals around zero; a higher value indicates that the model prediction errors are more concentrated. Peak position characterizes the symmetry of the residual distribution; a value closer to zero indicates smaller systematic bias. The MOE-XGB model exhibits higher peak density across all intensity measures, indicating a more concentrated residual distribution. Meanwhile, the peak position of MOE-XGB is closer to zero, indicating that the model does not exhibit significant systematic bias. These features are also intuitively reflected in Figure 6: the residual distribution of MOE-XGB (top row) is generally sharper and more symmetrical, whereas the GPR model (bottom row) presents flatter and more diffused characteristics.

In summary, the MOE-XGB model not only possesses higher prediction accuracy but also comprehensively outperforms the GPR model in terms of the concentration, symmetry, and stability of residual distribution, demonstrating its potential advantages in ground-motion parameter modeling.

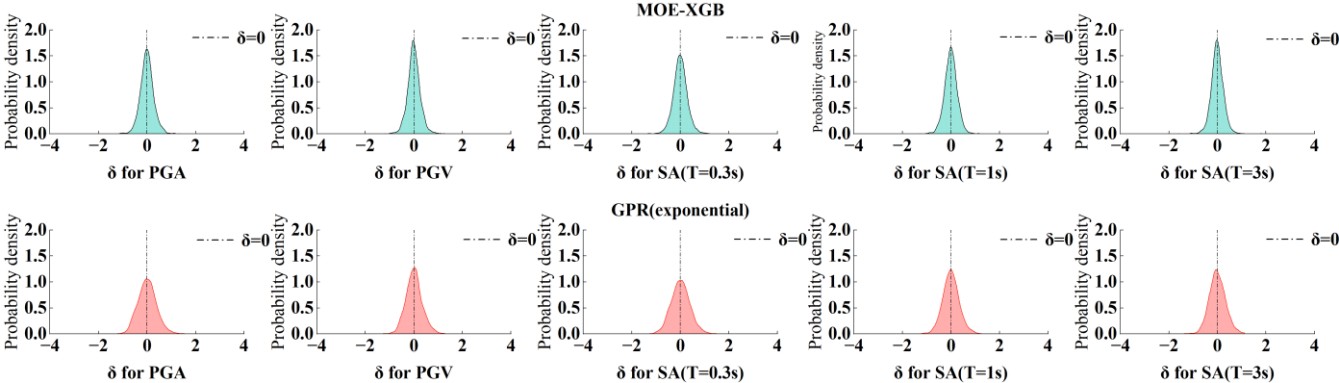



Figure 6. Comparison of residual probability density distributions for the MOE-XGB and GPR (exponential) models based
on the MORI dataset for PGA and different SA periods.

## 4.3 Comprehensive model evaluation

In this study, the proposed MOE-XGB model demonstrates significant advantages over the traditional GPR and Bindi 2017 models across multiple evaluation metrics. By comparing key metrics such as RMSE (root mean square error), $R$ (correlation coefficient), and Std Dev (standard deviation), MOE-XGB exhibits prominent performance in terms of prediction accuracy 295 and stability, as shown in Table 2 and Figure 7. First, regarding the RMSE metric, the error value for MOE-XGB is 0.2253, significantly lower than that of GPR (0.3633) and Bindi 2017 (0.4635). The RMSE of MOE-XGB is reduced by approximately 38.1 % compared to GPR and by approximately 51.4 % compared to Bindi 2017, indicating a clear advantage in prediction accuracy. Second, in terms of the $R$ value, MOE-XGB achieves 0.9569, far exceeding GPR (0.8991) and Bindi 2017 (0.8449). Compared to GPR, the correlation coefficient of MOE-XGB increased by approximately 6.4 %, and 300 compared to Bindi 2017, it increased by approximately 13.3 %, demonstrating that the model possesses stronger fitting capability and can better capture the variation patterns of ground-motion data. In the comparison of standard deviation (Std Dev), the value for MOE-XGB is 0.2654, which is a reduction of approximately 29.6 % and 41.0 % compared to GPR (0.3774) and Bindi 2017 (0.4498), respectively, further verifying the superiority of the MOE-XGB model in terms of stability and consistency.

Table 2. Comparison of metrics for different models.

| Metric | MOE-XGB | GPR | Bindi 2017 |
| --- | --- | --- | --- |
| Std Dev | 0.265 | 0.377 | 0.449 |
| RMSE | 0.225 | 0.363 | 0.463 |
| R | 0.956 | 0.899 | 0.844 |

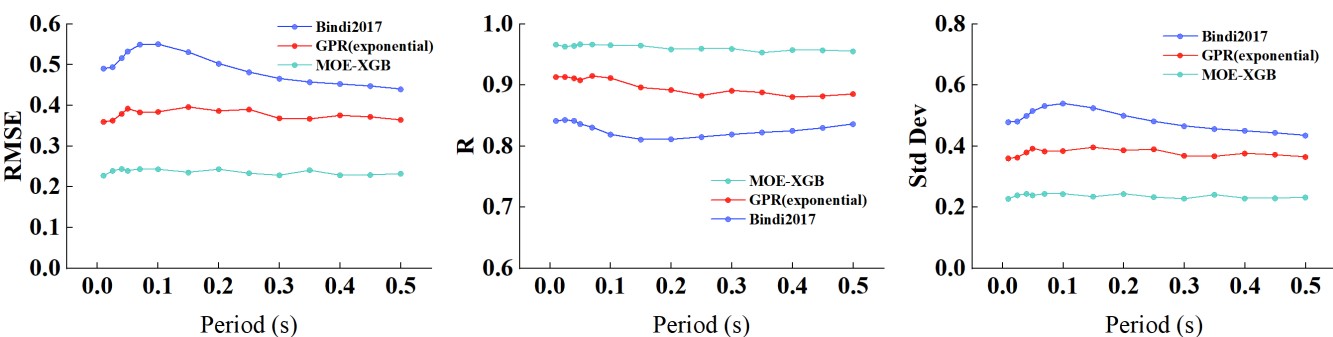

Figure 7. Comparison of performance metrics for different models (left: RMSE; middle: $R$; right: Std Dev).

Combined with the aforementioned metric comparison results, residual analysis further verifies the advantages of the MOE-XGB model in ground-motion prediction. Judging from the residual plots (see Figure 8) and the number of points with





absolute residuals greater than 1, the MOE-XGB model shows a significant improvement in prediction accuracy across all periods compared to the GPR and Bindi 2017 models. For the PGA period, the number of points with residuals greater than 1 for MOE-XGB is only 9, whereas for GPR and Bindi 2017, the numbers are 58 and 273, respectively, representing reductions of approximately 84.5 % and 96.7 %. For the SA ($T$=0.3 s) and SA ($T$=1 s) periods, the number of points with residuals greater than 1 for MOE-XGB decreased by 78.6 % and 94.8 % compared to GPR, and by approximately 91.5 % and 94.0 % compared to Bindi 2017. Similarly, for the SA ($T$=3 s) period, MOE-XGB exhibits a clear advantage, with 12 points having residuals greater than 1, representing reductions of approximately 75.5 % and 94.0 % compared to GPR (49 points) and Bindi 2017 (201 points), respectively. The significant reduction in residuals indicates that MOE-XGB can more accurately fit ground-motion data across different periods; particularly at short periods, its prediction results are more precise and stable compared to traditional models. The residual distribution in Figure 9 further demonstrates that the prediction accuracy of the MOE-XGB model is superior to the other two models, illustrating that MOE-XGB, based on the mixture of experts framework, possesses stronger adaptability and performance when processing complex ground-motion data.

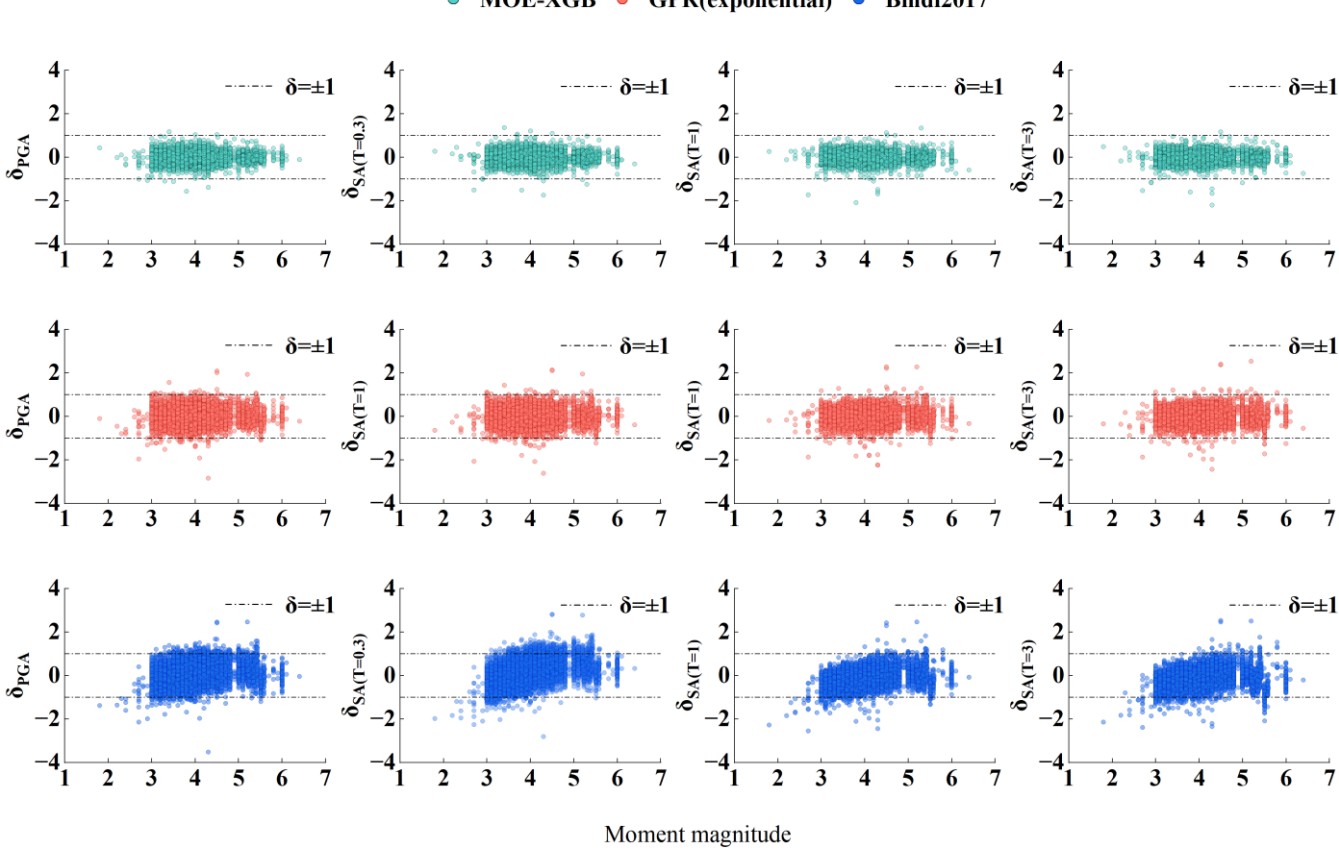

Figure 8. Variation of residuals with moment magnitude for different models based on the ITACA dataset for PGA and different SA periods.

Figure 9. Variation of residuals with epicentral distance for different models based on the ITACA dataset for PGA and

different SA periods.

To comprehensively evaluate the prediction accuracy of the model, we first conducted an intra-event residual analysis, as shown in Figure 10. Intra-event residuals mainly focus on the prediction errors of the model within the same seismic event, revealing performance differences across different periods, particularly regarding accuracy and stability. The analysis results show that the MOE-XGB model is significantly superior to GPR and Bindi 2017 in controlling intra-event residuals. In the

PGA period, the number of points with intra-event residuals greater than 1 or less than −1 for MOE-XGB is only 5, whereas for GPR and Bindi 2017, the numbers are 16 and 44 (of which 25 are greater than 1 and 19 are less than −1), respectively. MOE-XGB reduced the count by approximately 68.8 % compared to GPR and by approximately 88.6 % compared to Bindi 2017. In the SA ($T$ =0.3 s) period, the number of points with intra-event residuals greater than 1 or less than -1 for MOE-XGB is 2, compared to 19 for GPR and 57 for Bindi 2017; MOE-XGB reduced the count by approximately 89.5 %

compared to GPR and by approximately 96.5 % compared to Bindi 2017. In the SA ($T$ =1 s) period, the number of points for



MOE-XGB is 7, compared to 13 for GPR and 18 for Bindi 2017; MOE-XGB reduced the count by approximately 46.2 % compared to GPR and by approximately 61.1 % compared to Bindi 2017. In the SA ($T$ =3 s) period, the number of points for MOE-XGB is 6, compared to 12 for GPR and 13 for Bindi 2017; MOE-XGB reduced the count by approximately 50 % compared to GPR and by approximately 53.8 % compared to Bindi 2017.

To further examine the consistency and stability of the model across different seismic events, we conducted an inter-event residual analysis, as shown in Figure 11. Inter-event residuals mainly reflect the generalization capability of the model across multiple seismic events, especially its adaptability when facing different earthquake characteristics. In this analysis, MOE-XGB also demonstrated strong advantages, particularly in controlling the number of large residuals. In the PGA period, the number of points with inter-event residuals greater than 1 or less than −1 for MOE-XGB is only 4, which is a significant

reduction of approximately 71.4 % and 87.5 % compared to GPR (14 points) and Bindi 2017 (32 points), respectively. In the SA ($T$ =0.3 s) period, the number of points for MOE-XGB is 3, compared to 9 for GPR and 38 for Bindi 2017; MOE-XGB reduced the count by approximately 66.7 % compared to GPR and by approximately 92.1 % compared to Bindi 2017. In the SA ($T$ =1 s) period, the number of points for MOE-XGB is 2, compared to 9 for GPR and 42 for Bindi 2017; MOE-XGB reduced the count by approximately 77.8 % compared to GPR and by approximately 95.2 % compared to Bindi 2017. In the

SA ($T$ =3 s) period, the number of points for MOE-XGB is 5, compared to 11 for GPR and 51 for Bindi 2017; MOE-XGB reduced the count by approximately 54.5 % compared to GPR and by approximately 90.2 % compared to Bindi 2017.





Figure 10. Intra-event residual analysis for different models based on the ITACA dataset for PGA and different SA periods.







Figure 11. Inter-event residual analysis for different models based on the ITACA dataset for PGA and different SA periods.

In the comparison of residual probability density distributions, we further analyzed the performance differences between the

MOE-XGB model and the GPR and Bindi 2017 models by calculating peak density, skewness, kurtosis, and peak position. According to the data in the table, the peak density of MOE-XGB across various periods is consistently higher than that of GPR and Bindi 2017, indicating that the prediction residuals of MOE-XGB are more concentrated. Specifically, in the PGA period, the peak density of MOE-XGB is 2.1261, significantly higher than the 1.1771 of GPR and 0.8231 of Bindi 2017; in the 0.3 s period, the peak density of MOE-XGB is 2.1688, also higher than the 1.1172 of GPR and 0.8887 of Bindi 2017; in

the 1.0 s period, MOE-XGB reaches 2.3250, compared to 1.2555 for GPR and 1.0388 for Bindi 2017; and in the 3.0 s period, MOE-XGB is 2.2464, significantly higher than the 1.3374 of GPR and 1.0878 of Bindi 2017. Furthermore, the peak position of the MOE-XGB model is close to zero, indicating that its residual distribution is highly symmetrical with almost no systematic bias. For instance, the peak position of MOE-XGB is −0.0132 in the PGA period, −0.0061 in the 0.3 s period, −0.0066 in the 1.0 s period, and −0.0170 in the 3.0 s period; these results are all close to zero, indicating that the prediction

error of MOE-XGB is nearly zero and the residuals are symmetrical, demonstrating extremely high precision and reliability. In contrast, the peak positions of GPR and Bindi 2017 generally deviate from zero, suggesting the presence of systematic




bias in their predictions. particularly in the PGA period, the peak position is −0.0382 for GPR and 0.1796 for Bindi 2017, both exhibiting noticeable deviations. Therefore, the superiority of the MOE-XGB model lies in its more precise and stable prediction results; not only are the residuals concentrated around zero with small bias, but the volatility of prediction errors is

also significantly lower than that of other models, making it suitable for ground-motion prediction tasks requiring high precision.



Figure 12. Comparison of residual probability density distributions for different models based on the ITACA dataset for PGA and different SA periods.

Figure 13(a) and (b) display the prediction results for the same seismic event at different stations, estimated by the MOE-

XGB, GPR, and Bindi 2017 models, respectively. Selecting data from different stations for the same seismic event, rather than records from different seismic events, was primarily done to ensure that the response characteristics of the models to different stations are compared under the same seismic source conditions. This approach eliminates differences in source characteristics between seismic events, allowing for a more accurate analysis of the impact of different station conditions on prediction results, thereby evaluating the adaptability and prediction accuracy of the models under different ground-motion

propagation paths and local soil conditions.





In Figure 13(a) and (b), the prediction results of the MOE-XGB model show significant superiority compared to the actual recorded data, especially in the high- and medium-frequency period ranges. The MOE-XGB model provides prediction results closest to the recorded data across a large portion of the period range, particularly in the 0.1 to 1 s period range and the 1 to 2 s period interval. Compared to GPR and Bindi 2017, the predictions of MOE-XGB are more precise across most

period ranges, especially in low-frequency and high-frequency ground-motion responses. MOE-XGB effectively captures the ground-motion characteristics of stations either near to or far from the epicenter, providing more stable and precise predictions, which demonstrates its strong adaptability and universality. Whether in the high-frequency or low-frequency regions, the predictions of MOE-XGB exhibit high accuracy, indicating its distinct advantages in practical earthquake engineering applications, particularly in the prediction of structural responses for buildings and bridges.

It is worth noting that the recorded data in the figures come from the ITACA dataset. Since the update of the ITACA dataset concluded in 2022, this event represents the latest seismic event in the dataset. Furthermore, the seismic events used for this comparison were not included in the training, validation, or testing datasets, thereby excluding the possibility of data leakage. Through this setup, we can ensure that the model's prediction results are truly based on unknown data, further verifying that the MOE-XGB model can still provide accurate predictions when facing the latest ground-motion records that were not

involved in training.

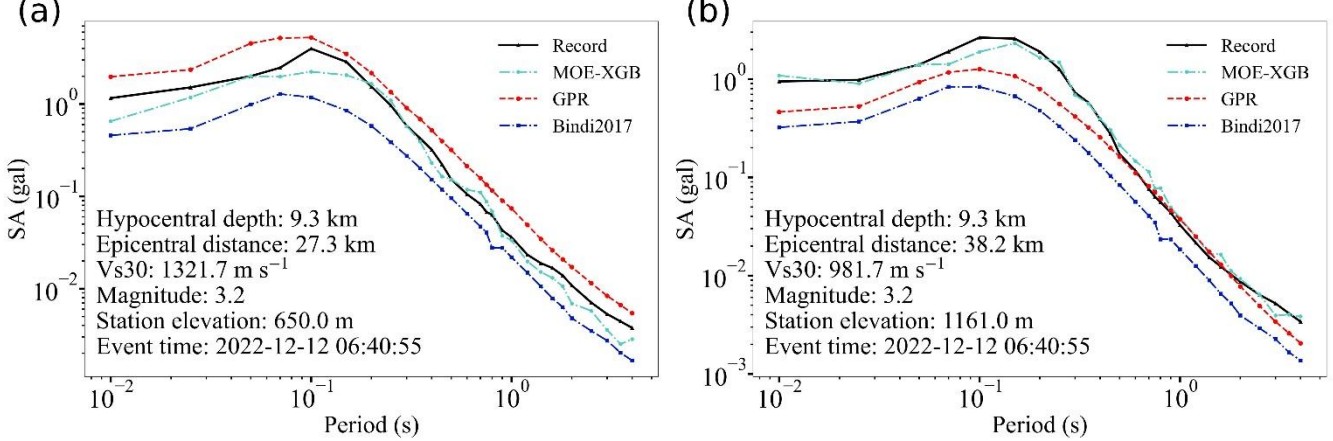

Figure 13. Comparison of predicted values and measured records for different mod

## 5 Conclusions

The MOE-XGB model proposed in this study demonstrates significant advantages in ground-motion prediction compared to the traditional GPR and Bindi 2017 models. Through multiple evaluation metrics, such as RMSE, the $R$ value, and standard

deviation (Std Dev), we verified the superiority of MOE-XGB in terms of prediction accuracy and stability. Particularly in the residual analysis of short and long periods, the MOE-XGB model exhibited more precise and consistent prediction results under different seismic event and station conditions. Compared with the GPR and Bindi 2017 models, the prediction



residuals of MOE-XGB are smaller, and across multiple periods, the model can better capture ground-motion characteristics, displaying strong adaptability and stability.

Furthermore, this study further verified the generalization capability of the MOE-XGB model in various ground-motion scenarios through intra-event and inter-event residual analyses. The intra-event residual analysis indicates that MOE-XGB provides more precise prediction results under different station conditions for the same seismic event, excluding the risk of data leakage and ensuring the independence and reliability of model evaluation. The inter-event residual analysis further indicates that MOE-XGB can effectively handle variations between different seismic events, demonstrating stronger model

consistency and stability.

By validating on the latest ground-motion records in the ITACA dataset, we ensured that the model still possesses high prediction accuracy when facing unknown data. This design avoids the limitations of the dataset, particularly against the backdrop where the traditional dataset is updated only until 2022, MOE-XGB demonstrated good adaptability and performance when facing the latest seismic events.

Overall, the MOE-XGB model exhibits significant advantages by virtue of its excellent predictive capability, stability, and high accuracy, especially in practical engineering applications in the field of ground-motion prediction. Furthermore, high-precision ground-motion prediction models form the foundation for Probabilistic Seismic Hazard Analysis (PSHA) (Weatherill et al., 2024; Ullah et al., 2025) and serve as a cornerstone for assessing future seismic risks. Future research can further explore the application of MOE-XGB in different seismic scenarios, especially in complex ground-motion

environments and broader seismically active regions, to verify its applicability and scalability in a wider range of engineering problems.

**Code and data availability**

The MORI ground-motion dataset was downloaded from https://data.ingv.it/dataset/404#additional-metadata (last access: 8 December 2025). The ITACA dataset was downloaded from https://itaca.mi.ingv.it/ItacaNet_40/#/home (last access: 12

December 2025). The model code is available from the corresponding author upon request.

**Author contributions**

JD designed the model and prepared the original draft. ZW supervised the project and secured funding. DZ and XW contributed to data curation. JW and ZL performed the formal analysis and validation. ZDW and JX assisted with visualization. All authors contributed to the review and editing of the manuscript.



**Competing interests**

The contact author has declared that none of the authors has any competing interests.

**Disclaimer**

Copernicus Publications remains neutral with regard to jurisdictional claims made in the text, published maps, institutional affiliations, or any other geographical representation in this paper.

**Financial support**

This research has been supported by the Scientific Research Fund of the Institute of Engineering Mechanics, China Earthquake Administration (grant no. 2023A01) and the National Natural Science Foundation of China (grant nos. 52378544 and 52378543).

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
