# Peer review of "A ground motion prediction model for the Italian region based on a mixture of experts framework"

_EGUsphere, 2025_

## Author Comment (AC1)

**Author Comment (AC)**
**Response to Referee #1 (All comments)**

*Manuscript: "A ground motion prediction model for the Italian region based on a mixture of experts framework"*

**Referee #1 – Comment 1 (Validation strategy and event-wise independence)**

The referee expresses concern that random record-wise data splits may allow recordings from the same earthquake to appear in both training and testing subsets, causing event-level information leakage and overly optimistic performance estimates. The referee requests an explicit event-wise validation/testing design and recomputation of all reported metrics and residual diagnostics under event-wise independence.

**Author response (intended revisions)**

We thank the referee for raising this critical methodological point. We agree that ground-motion recordings from the same earthquake are correlated through shared source, path, and rupture characteristics, and that a purely random record-wise split can introduce event-level information leakage if not carefully controlled.

In the discussion paper, our initial use of a random split was primarily motivated by practical coverage considerations (to maintain broad and comparable coverage across magnitude, distance, and site-condition ranges in the development subsets). However, we acknowledge that this choice does not, by itself, provide a sufficiently strict basis for claims about generalization to unseen earthquake events. In the revised manuscript, we will therefore (i) document the validation/testing strategy more explicitly and (ii) revise wording in the abstract, results, and conclusions so that any statements regarding "independent events" are supported by event-wise independent evaluation.

**Planned evaluation redesign (event-wise independence)**

To address the referee's request while also preserving strict comparability with the benchmark study used in our manuscript, we will report results under two complementary experimental tracks corresponding to the two datasets considered in the paper:

**Track A: Benchmark dataset aligned with the published GPR comparison study**

- Objective: ensure an apples-to-apples comparison with the benchmark study by reproducing its data partitioning and evaluation protocol as closely as possible.
- Training/validation: we will follow the benchmark study's 5-fold cross-validation procedure on its training/validation pool to mirror the reported workflow and hyperparameter selection strategy.

- Independent testing: we will retain the 2016-10-30 Central Italy Mw 6.5 event (241 station recordings) as the held-out test event, fully excluded from all model development steps.
- Reporting: on this held-out event, we will recompute and report key generalization metrics (e.g., RMSE and correlation) and residual diagnostics (e.g., residual distributions and event/within-event dispersion) for our MoE model and the benchmark GPR model (and any baseline GMPEs used in the benchmark study, where applicable).

Track B: Our expanded official dataset (primary basis for assessing robustness on a larger dataset)

- Data splitting: we will adopt a record-wise partitioning strategy consistent with the benchmark framework, designed to preserve comparable magnitude–distance–site-condition coverage between development and test subsets.
- Development (validation): we will use 5-fold cross-validation on the development pool to set any hyperparameters and finalize the model configuration, without using any held-out test data.
- Final training: after fixing the final configuration, we will retrain the MoE model on the full development pool.
- Independent testing: we will evaluate the final model on a held-out test subset that is not used in any training or validation step, and we will recompute performance metrics and residual diagnostics under this fixed protocol.
- Baselines: under the same train/validation/test protocol, we will compare MoE against the GPR baseline and selected conventional GMPE baselines in a consistent manner (same IM definition, preprocessing, and residual definition).

Across both tracks, we will ensure that hyperparameter choices are made using only development data, and we will not adjust models after observing test-set performance. All figures/tables and statements in the manuscript that summarize predictive performance will be updated accordingly, and any conclusions about generalization strength will be based on the fixed held-out test results described above.

**Referee #1 – Comment 2 (Interpretation of aleatory variability and residual reduction)**

The referee questions our interpretation of the reduced residual dispersion (σ) reported for the MoE model and notes that, for flexible ML models, variance reduction may reflect methodological artifacts rather than a physically meaningful reduction of aleatory variability. The referee requests that we (i) clarify how residual components are estimated for complex ML architectures, (ii) benchmark the inferred variability against published GMPE variability models (σ, τ, ϕ), and (iii) discuss physical plausibility versus potential artifacts.

**Author response (intended revisions)**

We thank the referee for this important and nuanced comment. We agree that, in the context of ground-motion modeling, a reduction of overall residual dispersion in a flexible ML model should not be interpreted automatically as a reduction of aleatory variability in the GMPE sense. We also agree that reductions in residual variance can, in principle, arise from methodological factors (e.g., model flexibility, implicit smoothing, or event-specific pattern exploitation) and therefore require careful evaluation and benchmarking against established GMPE practice.

**Clarification of terminology and scope**

In the discussion paper, our reported "σ reduction" primarily referred to a decrease in the overall residual dispersion computed from residuals of the form r = ln(Y_obs) − ln(Y_pred). In the revised manuscript, we will make explicit that this overall dispersion is not necessarily equivalent to the aleatory variability as commonly characterized in GMPEs through the decomposition into between-event and within-event components. We will therefore avoid language that could be read as claiming a direct physical reduction of aleatory variability solely on the basis of smaller overall residual scatter.

**Planned additions and analyses (to address items i–iii)**

- **(i) Residual-component estimation for ML predictions**
  We will add a dedicated methodological description explaining how variability components will be estimated from ML-based residuals under an independent testing framework. Specifically, we will follow the standard event-term decomposition commonly used in GMPE residual analysis, i.e., r_ij = η_i + ε_ij, where η_i represents the event term (between-event component) and ε_ij the within-event residual. In practice, we will estimate η_i as the event-mean residual and compute within-event residuals by subtracting η_i from record-level residuals; the corresponding dispersions will be summarized through τ (between-event), ϕ (within-event), and σ = √(τ² + ϕ²). We will clearly document definitions, computation steps, and the evaluation split used, to ensure transparency and reproducibility.

- **(ii) Benchmarking against published GMPE variability models (σ, τ, ϕ)**
  We will benchmark the inferred variability levels of the MoE model against the published variability models of the reference GMPEs already included in our manuscript/baseline comparisons (Italian and/or pan-European models as used in the paper). The revised

manuscript will present side-by-side comparisons of σ, τ, and ϕ (derived under the same residual definition and evaluation framework) to assess whether the resulting variability levels are consistent with established ranges and physically plausible.

- **(iii) Physical meaning versus methodological artifacts**

  We will expand the discussion to interpret any observed reductions in overall dispersion (and/or in τ and ϕ) in a more cautious and physically informed manner. In particular, we will explicitly consider two complementary explanations: (a) potential physically meaningful improvements, e.g., improved representation of source/path/site proxies and/or better partitioning of systematic effects through the MoE framework that may reduce unmodeled structure in residuals; and (b) potential methodological contributors, e.g., smoothing effects or limited-sample artifacts that could reduce apparent scatter without corresponding physical justification. To support this discussion, we will also add residual diagnostics (e.g., distributional checks and trends with key predictors such as magnitude, distance, and site proxies) to verify that reduced dispersion is not masking systematic biases.

**Planned revisions to manuscript wording**

Finally, we will revise the abstract, results, and conclusions to ensure that statements regarding "aleatory variability reduction" are appropriately qualified and are supported by the above decomposition and GMPE benchmarking. Any claims about variability will be framed in terms of (a) overall residual dispersion versus (b) GMPE-style variability components (σ, τ, ϕ), and will be grounded in the independent evaluation framework adopted in the revised manuscript.

**Referee #1 – Comment 3 (Definition and documentation of predictors)**

The referee notes that the manuscript does not clearly and consistently document the full set of predictors used in the MoE-XGB model. While some variables are mentioned in different sections, an unambiguous and complete list of input features (with definitions, sources, and preprocessing) is missing, which limits reproducibility and makes it difficult to assess physical consistency.

**Author response (intended revisions)**

We thank the referee for highlighting this reproducibility issue. We agree that the current description of predictors is dispersed across sections and does not provide a single, complete, and unambiguous inventory of the model inputs. This can hinder readers' ability to reproduce the workflow and to interpret the predictors in a physically consistent manner.

**Planned revisions**

- **Predictor summary table (complete feature list).**
  In the revised manuscript, we will add a dedicated table that lists all predictors used by the MoE-XGB model in a clear and consistent manner. For each predictor, the table will provide (i) the variable name (consistent with the model implementation), (ii) a brief definition and unit, (iii) the data source (as used in the manuscript), and (iv) a concise note on the main preprocessing step(s) applied (e.g., scaling/transformations and handling of missing values). This table will serve as the definitive reference for the model's input specification.

- **Feature importance (relative roles).**
  To address the referee's request for discussing the relative roles of predictors, we will add a robust feature-importance analysis (e.g., model-based importance and/or permutation-based importance, as appropriate for the adopted learners). This analysis will be presented as a quantitative aid to interpretability, and we will explicitly clarify that it is intended to support transparency rather than to imply direct physical causality.

**Referee #1 – Comment 4 (Lack of interpretability analysis)**

The referee notes that, given the complexity of the proposed MoE-XGB framework and the strong performance claims, the manuscript would benefit from a systematic interpretability analysis. In particular, the referee requests explicit global interpretability results (e.g., SHAP-based importance such as beeswarm/summary plots) to clarify which predictors dominate the predictions across different intensity measures and spectral periods.

**Author response (intended revisions)**

We thank the referee for this constructive suggestion. We agree that, for a complex framework such as MoE-XGB, reporting predictive accuracy alone is not sufficient to fully support scientific interpretation. Adding a dedicated interpretability analysis will improve transparency, strengthen the physical consistency discussion, and help distinguish physically meaningful patterns from purely data-driven behavior.

**Planned revisions**

- **SHAP-based global interpretability.**
  In the revised manuscript, we will include an explicit SHAP-based global interpretability analysis for the MoE-XGB model. We will present global importance results using standard SHAP summary visualizations (e.g., beeswarm/summary plots) to identify the predictors that most strongly control the model outputs.

- **Interpretability across IMs and periods.**
  We will report SHAP global importance separately for different intensity measures and spectral periods considered in the manuscript (e.g., PGA/PGV and Sa at multiple periods).

**Referee #1 – Comment 5 (Dataset definition and currency: ITACA)**

The referee notes that the manuscript refers to the ITACA dataset but does not clearly specify which version of the ITACAext flatfile is used. Given that recent releases (e.g., ITACAext flatfile 2.0; Lanzano et al., 2024) include updated metadata and intensity measures, the referee requests that we explicitly state the exact dataset version and provide a precise data citation to ensure transparency and reproducibility.

**Author response (intended revisions)**

We thank the referee for raising this important transparency and reproducibility issue. We agree that, for strong-motion flatfiles, versioning can affect metadata completeness and intensity-measure definitions, and therefore the exact dataset release must be reported explicitly.

**Planned revisions**

- **Explicit dataset version statement.**
  In the revised manuscript, we will explicitly state that we use the ITACAext flatfile version 2.0 and cite the corresponding reference (Lanzano et al., 2024). We will also specify the dataset access information (including the download/access date) in the data section to make the dataset provenance fully traceable.

- **Consistent data citation and documentation.**
  We will ensure that the ITACAext v2.0 citation is consistently included wherever the dataset is introduced or used (data description and any data-availability statement), and that the manuscript clearly links all reported analyses and results to this specific dataset release.

**Referee #1 – Comment 6 (Overstatement of applicability to seismic hazard)**

The referee notes that the manuscript repeatedly suggests applicability to seismic hazard assessment and PSHA, while no hazard-oriented application is demonstrated and key aspects (e.g., spatial correlation, rupture geometry, and PSHA workflow integration) are not addressed. The referee requests that we either substantially tone down these claims or explicitly demonstrate PSHA integration.

**Author response (intended revisions)**

We thank the referee for this important comment and agree that our current wording may overstate PSHA readiness relative to what is explicitly demonstrated in the discussion paper. In the revised manuscript, we will ensure that all statements about seismic-hazard applicability are aligned with the analyses presented and are framed with appropriate scope and limitations.

**Planned revisions**

- **Tone down and scope PSHA-related claims.**
  We will systematically revise the abstract, introduction, discussion, and conclusions to avoid strong statements implying direct, end-to-end PSHA application. Where PSHA is mentioned, we will rephrase it as a potential application and clearly state that the primary contribution of this work is the development and validation of a regional ground-motion prediction model, while a full hazard application requires additional components and assumptions.

- **Document a concrete integration pathway (methodological description).**
  To support the (more cautious) PSHA-related statements, we will add a concise methodological description outlining how the proposed MoE-XGB model can be packaged and used as a GMPE/GSIM within a PSHA workflow. This description will clarify: (i) required inputs and outputs (median prediction and an associated variability specification suitable for PSHA use), (ii) how the model would be called within a standard PSHA engine (e.g., OpenQuake) at the workflow level, and (iii) which additional elements are outside the scope of the present paper but are necessary for a complete hazard implementation (e.g., rupture geometry and distance metrics selection, spatial correlation models, and interfaces with source models/logic trees).

**Referee #1 – Comment 7 (Minor comments and technical issues)**

We thank the referee for these helpful editorial and technical suggestions. We agree that addressing them will improve clarity, reduce redundancy, and strengthen the readability and engineering interpretability of the manuscript.

**Planned revisions**

- **Reduce redundancy in the Results section and consolidate figures.**
  We agree that several figures convey overlapping information. In the revised manuscript, we will streamline the Results section by consolidating highly similar figures, retaining only those that provide distinct insight, and moving secondary or repetitive material to the Supplement (or removing it where appropriate). We will also revise the associated text and captions to clarify the specific purpose and added value of each remaining figure.

- **Clarify the meaning of performance averages across periods/IMs.**
  Where performance metrics are averaged across spectral periods and/or intensity measures, we will either (i) provide a clear justification for why such averaging is informative (as a high-level summary only), or (ii) avoid presenting averaged values as primary engineering evidence. In all cases, we will emphasize period-specific and IM-specific results as the main basis for interpretation.

- **Replace potentially misleading terminology ("MORI dataset").**
  We agree that the term "MORI dataset" may be confusing. In the revised manuscript, we will replace it with a clearer description (e.g., a compiled Italian strong-motion dataset) and will explicitly state the contributing data sources and compilation/selection criteria in the data section to avoid ambiguity.

- **Correct typographical/grammatical issues and improve consistency.**
  We will perform a thorough language and technical consistency check throughout the manuscript, correcting typographical and grammatical errors and harmonizing terminology, symbols, abbreviations, figure/table references, and units.

---

## Author Comment (AC2)

**Author Comment (AC)**
**Response to Referee #1 (All comments)**

*Manuscript: "A ground motion prediction model for the Italian region based on a mixture of experts framework"*

**Referee #1 – Follow-up on Comment 1 (Validation strategy and event-wise independence)**

The referee acknowledges that most issues raised in the initial review have been satisfactorily addressed, but emphasizes that the validation strategy still requires a fully event-wise independent evaluation. Specifically, the referee clarifies that strict event-wise validation in ground-motion modeling should be implemented through leave-one-event-out (LOEO) or an equivalent grouped-by-event cross-validation scheme, where the model is trained on all events except the excluded event(s) and performance metrics/residual statistics are computed exclusively on the fully excluded event(s). Final metrics should then be reported as averages over all cross-validation iterations. The referee notes that (i) Track A (single held-out event) is informative as a case study/benchmark comparison but not a robust unseen-event generalization assessment, and (ii) Track B, as previously formulated, does not unambiguously enforce strict event-wise independence.

**Author response**

We thank the referee for this additional clarification and for recognizing the improvements in our planned revisions. We fully agree that this point concerns experimental design rather than wording, and that a robust assessment of generalization to unseen earthquakes requires at least one evaluation that enforces strict event-wise independence in the sense defined by the referee. In the revised manuscript, we will therefore incorporate an explicit grouped-by-event cross-validation evaluation and report all key performance and residual-variability metrics under this event-wise independent framework.

At the same time, we will maintain our benchmark-oriented evaluation (previous Track A) strictly for its intended purpose—namely, direct comparability with the reference study used in our manuscript—while clearly delimiting its interpretive scope.

**Planned revisions (updated evaluation design)**

**(1) Track A retained as benchmark-aligned case study (scope clarified)**

- Purpose: We will retain the single held-out earthquake evaluation to preserve strict comparability with the benchmark study used in our manuscript.

- Scope: We will explicitly frame Track A as a benchmark-aligned case study, informative for reproducibility and direct comparison, but not as a standalone, robust quantification of generalization to unseen earthquakes.

**(2) Track B revised to strict event-wise independence via grouped-by-event cross-validation (new primary generalization assessment)**

We will replace the previous record-wise evaluation in Track B with an explicit **grouped-by-event cross-validation scheme**, implemented with **group = event_id**:

- Event-wise split rule: All splits will be performed by event, such that recordings from the same earthquake cannot appear in both training and testing data in any iteration.
- Cross-validation protocol: For each fold, the model will be trained on all events outside the held-out fold, and all residuals/metrics will be computed exclusively on the fully excluded event-fold.
- Reported metrics under event-wise CV: We will recompute and report the full set of performance and residual-variability statistics under this framework, including (as applicable) RMSE, correlation/bias diagnostics, residual dispersion measures, and variability decomposition metrics ($\tau$, $\phi$, $\sigma$), computed fold-wise on excluded events.
- Aggregation across folds: Final reported values will be summarized as averages across folds (and we will also report the dispersion across folds, to convey stability/uncertainty of the estimates).
- Interpretation: Claims regarding generalization to unseen earthquakes and the interpretation of residual variability will be based primarily on this event-wise cross-validation evaluation.

**Concluding note**

We believe that adding this explicit grouped-by-event cross-validation evaluation will directly address the referee's methodological requirement for strict event-wise independence, strengthen the robustness of our reported predictive performance and variability analyses, and provide a sound basis for interpreting improvements relative to baseline models.

---

## Author Comment (AC3)

**Author Comment (AC)**
**Response to Referee #1 (All comments)**

*Manuscript: "A ground motion prediction model for the Italian region based on a mixture of experts framework"*

**Referee #1 – Additional follow-up (Sensitivity to validation design under identical conditions)**

The referee acknowledges our commitment to implementing strict grouped-by-event cross-validation, and further emphasizes that the methodological robustness would be strengthened by explicitly quantifying how performance and residual-variability metrics change relative to the previously adopted record-wise evaluation, under identical dataset and predictor conditions. This comparison would improve transparency and allow readers to assess the sensitivity of the results to the validation design.

**Author response**

We thank the referee for this further clarification and fully agree with the underlying point. Beyond incorporating a strict grouped-by-event cross-validation scheme, it is important to explicitly quantify the sensitivity of the reported results to the choice of validation design under identical dataset and predictor conditions.

Accordingly, in the revised manuscript we will include an additional controlled comparison in which the dataset and model input configuration are kept unchanged and only the validation strategy is varied (record-wise evaluation versus grouped-by-event cross-validation). We will present this comparison as a complementary sensitivity analysis, reporting the resulting differences in evaluation outcomes side-by-side. We believe this addition will improve transparency, facilitate a more robust interpretation of the model performance and residual-variability findings, and further strengthen the scientific value of the study.

**Concluding note**

We again thank the referee for the constructive guidance. We believe that the above addition, together with the strict grouped-by-event evaluation already planned, will directly address the remaining methodological concerns and will provide a clearer, more robust basis for the manuscript's main conclusions.